# Long-Term Risk of Colectomy in Patients with Severe Ulcerative Colitis Responding to Intravenous Corticosteroids or Infliximab

**DOI:** 10.3390/jcm11061679

**Published:** 2022-03-18

**Authors:** Elena De Cristofaro, Silvia Salvatori, Irene Marafini, Francesca Zorzi, Norma Alfieri, Martina Musumeci, Emma Calabrese, Giovanni Monteleone

**Affiliations:** Department of Systems Medicine, University of Rome “Tor Vergata”, 00133 Rome, Italy; elena_decr@hotmail.it (E.D.C.); silviasalvatori23@gmail.com (S.S.); irene.marafini@gmail.com (I.M.); fra.zorzi@yahoo.it (F.Z.); norma.alfieri@outlook.com (N.A.); martinamusu92@gmail.com (M.M.); emma.calabrese@uniroma2.it (E.C.)

**Keywords:** inflammatory bowel disease, acute severe UC, steroids, infliximab, colectomy

## Abstract

Background and aims: Intravenous corticosteroids (IVCS) and rescue therapy with infliximab (IFX) are useful for managing patients with acute severe ulcerative colitis (ASUC). However, nearly one fifth of responders undergo colectomy. Predictive factors of colectomy in this subset of patients are not fully known. We retrospectively examined the long-term risk and the predictors of colectomy in ASUC patients achieving clinical remission following treatment with IVCS or IFX. Patients and methods: Clinical and demographic characteristics were evaluated in consecutive ASUC patients who were admitted to the “Tor Vergata University” hospital between 2010 and 2020 and responded to IVCS or IFX. A multivariate logistic regression model was constructed to identify independent predictors of colectomy. Results: A total of 116 ASUC patients responding to IVCS (98 patients) or IFX (18 patients) were followed up for a median of 46 months. After discharge, 29 patients (25%) underwent colectomy. Multivariate analysis showed that a serum albumin level <3 g/dL and colonic dilation >5.5 cm on admission were independent predictors of colectomy (OR: 6.9, 95% CI: 2.08–22.8, and OR 8.5, 95% CI: 1.23–58.3, respectively). Patients with both these factors had a risk of colectomy 13 times greater than those with no risk factor. Conclusions: A low serum albumin level and colonic dilation are risk factors of long-term colectomy in ASUC patients responding to IVCS or IFX.

## 1. Introduction

During their natural history, nearly one fourth of patients with ulcerative colitis (UC) experience acute severe clinical flare-ups defined by the Truelove and Witts criteria [1,2]. Intravenous corticosteroids (IVCS) are useful for managing hospitalized patients with acute severe UC (ASUC) [2,3]. Second-line therapy with cyclosporine or infliximab (IFX) is a rescue treatment for patients who do not respond to IVCS after 3–5 days [4]. Although these later approaches have markedly improved the prognosis of patients with IVCS-refractory UC, mainly in terms of the colectomy rate, urgent colectomy is still required in more than one fourth of patients [5]. Several predictors of colectomy have been identified in these subsets of ASUC patients. These include clinical characteristics (i.e., sex, stool frequency, previous exposure to biological therapies, and steroid dependence), biological parameters (i.e., albumin, C-reactive protein (CRP), and hemoglobin levels), and endoscopic and X-ray findings (i.e., “mucosal islands” and colonic dilatation) [5,6,7,8]. Superinfections, such as Clostridium Difficile and cytomegalovirus, are further prognostic factors [9,10].

In ASUC patients who respond to IVCS or IFX during their index admission, the colectomy rates tend to increase after a second hospital admission for ASUC [11]. In particular, patients who do not completely respond to steroid treatment have a 50% chance of requiring a colectomy within 1 year of discharge, which increases up to 70% at 5 years [12]. A recent scoring system was developed with the goal to identify patients at low risk vs. high risk for colectomy within 1 year of hospitalization for ASUC. Previous exposure to TNF antagonists or thiopurines, Clostridium difficile infection, and serum levels of CRP and albumin were identified as predictive factors for colectomy and included in the risk score. However, in this study, radiological and endoscopic criteria were lacking, and the time of follow-up after hospitalization was short [13]. Indeed, limited data are available about the predictive factors of long-term colectomy in ASUC patients who have a complete clinical response to medical therapy. In this study, we retrospectively examined the long-term risk of colectomy in patients with ASUC after the induction of clinical remission by IVCS or IFX and evaluated the predictors of colectomy.

## 2. Materials and Methods

### 2.1. Study Population and Data Collection

This retrospective study included all patients with an established UC diagnosis who were admitted to the “Tor Vergata University” hospital between 2010 and 2020 for severe relapse and successfully treated with IVCS or second-line therapy with IFX. UC clinical activity was defined according to the modified Truelove and Witts criteria [1]. The patients were excluded if they had a mild or moderate UC flare-up, evidence of intestinal superinfection with cytomegalovirus, Clostridium difficile, or other intestinal pathogens, a previous diagnosis of Crohn’s disease or unclassified inflammatory bowel disease (IBD), or if there was a failure of IVCS or IFX treatment, leading to emergency surgery.

Patients were recruited from the standardized hospital in-patient diagnostic electronic dataset by searching for the International Classification of Diseases (ICD-9) codes (556.0, 556.3, 556.6, 556.8, 556.9). Demographic and clinical characteristics were collected from medical records and included sex, age, smoking habits, medical history, disease extent (according to the Montreal classification [14]), date of diagnosis, and the dates of admission and discharge. Clinical disease activity was assessed at baseline using the partial Mayo Clinic score; laboratory activity was assessed using CRP (mg/L) and hemoglobin (g/dL) values recorded according to the Truelove and Witts criteria (>30 mg/L and <10.5 g/dL, respectively), and albumin levels (<3 g/dL). At admission, the body mass index (BMI) was calculated, and a value <18.5 kg/m^2^ was defined as the index of malnutrition according to the ESPEN diagnostic criteria [15]. Abnormal X-ray findings, including the presence of colonic dilation (>5.5 cm), were assessed on the first day of hospitalization.

Intensive medical therapy consisted of intravenous methylprednisolone (60 mg per day) with fluid and electrolyte resuscitation and thromboprophylaxis with subcutaneous low-molecular-weight heparin. Response to IVCS was defined as a resolution of clinical symptoms with ≤3 stools per day without visible bleeding. Refractoriness to IVCS was defined according to clinical and biochemical criteria on days 3 and 5. Rescue therapy was considered in patients with no symptomatic improvement following IVCS by day 5. All these patients were treated with IFX (5 mg/kg).

After discharge, the patients were evaluated for disease activity, treatment (including the need for steroids or biological therapies), and need for re-hospitalization, until the end of the follow-up corresponding to their last visit or colectomy. Data were obtained from follow-up outpatient records. The main endpoint was colectomy at any time during the follow-up. The study protocol conformed to the ethical guidelines of the 1975 Declaration of Helsinki as reflected in the a priori approval by the institution’s human research committee. No informed consent was required because of the retrospective nature of the study.

### 2.2. Statistical Analysis

All statistical analyses were performed using GraphPad Prism (Version 9.0) (GraphPad Software, Inc., San Diego, CA, USA). Qualitative data were expressed as the number and proportion (%), and quantitative data were expressed as the median (range) and mean ± standard deviation (SD). Patients’ characteristics were compared using the χ^2^ or Fisher exact test for the categorical variables, and with the Mann–Whitney test for continuous variables. A logistic regression was performed, and the parameters with *p* < 0.05 in the univariate analysis were used to perform a multivariate logistic regression analysis to determine their influence on colectomy. The results of the logistic regression analysis were expressed using odds ratios (ORs) and 95% confidence intervals (CIs) with the *p* values. To evaluate the predictors, receiver operating characteristic (ROC) curves were plotted. The colectomy-free survival was calculated using the Kaplan–Meier method.

## 3. Results

### 3.1. Study Population and Characteristics of Patients

Of 125 patients hospitalized for ASUCS, 9 underwent urgent colectomy for refractoriness to IVCS and IFX. The remaining 116 patients were included in the study; a total of 98 patients (85%) responded clinically to IVCS, and 18 patients (15%) achieved clinical remission after rescue therapy with IFX. During a median follow-up of 46 months (range 1–159), 29 out of 116 patients (25%) underwent colectomy. The median time of colectomy was 3 months (range 1–92), and 19 out of 29 patients (65%) underwent surgery within the first year after discharge. Table 1 shows differences between patients in the colectomy group and those in the non-colectomy group. The female gender, history of steroid dependence, and previous exposure to TNFs were more frequent in the colectomy group. Moreover, at admission, the partial Mayo Clinic scores and serum CRP levels were significantly greater, and albumin levels were significantly lower in patients undergoing colectomy as compared to the control group. The assessment of radiological findings showed that colonic dilation was greater in the colectomy group as compared to the control group. During the follow-up period, the numbers of patients who needed re-hospitalization, steroids, and biologics were higher in the colectomy group as compared to those seen in the non-colectomy group (Figure 1).

### 3.2. Predictors of Colectomy

Univariate analysis showed that female patients (OR: 2.914, 95% CI 1.21–6.97; *p* = 0.01) and those with a history of steroid dependency (OR: 3.224, 95% CI 1.36–7.66; *p* = 0.007) and previous exposure to anti-TNF-α therapy (OR: 2.808, 95% CI 1.07–7.34; *p* = 0.04) had a statistically higher risk of colectomy. Additionally, patients with serum CRP > 30 mg/L (OR: 2.92, 95% CI 1.01 to 8.43; *p* = 0.034) and albumin < 3 g/dL (OR 8.74, 95% CI 3.20–23.8; *p* < 0.0001) had a statistically higher risk of colectomy. Similarly, colonic dilation > 5.5 cm was significantly associated with colectomy (OR 7.58, 95% CI 2.06–27.87; *p* = 0.0016). Multivariate analysis showed that the serum albumin level and colonic dilation were independent predictors of colectomy (OR: 6.89, 95% CI: 2.08–22.8; *p* = 0.002, and OR 8.47, 95% CI: 1.23–58.3; *p* = 0.03, respectively) (Table 2). To establish which cut-off of these variables predicts the risk of colectomy, ROC curves were plotted for the serum albumin levels at admission for all 116 patients. The area under the curve, sensitivity, and specificity were 0.76, 73%, and 76%, respectively. The cut-off value for serum albumin was ≤2.8 g/dL (Figure 2). The same analysis was performed for colonic dilation. The area under the curve, sensitivity, and specificity were 0.83, 80%, and 73%, respectively, with a cut-off for colonic dilation of >4 cm (Figure 3).

### 3.3. Colectomy-Free Survival during Follow-Up

A colectomy-free survival curve was developed based on the absence or presence of one or two of the previously identified predictors. A total of 56 patients (48%) had no risk factor, 44 patients (38%) had one risk factor, and 16 patients (14%) had two risk factors. Forty-two patients reached 60 months of follow-up. At 5 years, the rates of colectomy-free survival with zero, one, and two risk factors were 93%, 76%, and 14%, respectively (Figure 4). Furthermore, the patients with albumin levels below 2.8 gr/dL and colonic dilation above 4 cm had a risk of colectomy 13 (95 CI%: 2.98 to 61.07) and 9.6 (95 CI%: 2.97 to 31.14) times greater than that documented in patients with no risk factor, or patients with only one risk factor, respectively. Patients with only one risk factor had a risk of colectomy four (95 CI%: 1.65 to 10.3) times greater than that seen in patients with no risk factor.

## 4. Discussion

Achieving complete clinical remission during the index hospital admission improves the long-term outcomes of ASUC patients [11]. Nonetheless, some of these patients undergo colectomy in the short- and long-term follow-up, clearly indicating the need for simple clinical and laboratory criteria that will predict outcomes and help clinicians to better manage the UC course following discharge. Many authors have attempted to identify clinical and biochemical predictive factors of colectomy. However, these studies were either aimed at identifying predictors of therapeutic failure during hospitalization for ASUC or involved patients with moderate to severe UC, with no distinction between responders and non-responders after hospitalization [16,17,18,19]. 

The present study was undertaken to assess the risk of colectomy in the long-term follow-up of ASUC patients achieving clinical remission after treatment with IVCS or IFX. Our data indicate that one fourth of patients underwent colectomy, thus confirming and expanding on previous results showing that the risk of colectomy in patients hospitalized for ASUC is about 20% after discharge [11,20,21,22,23]. 

Comparison of clinical and demographic characteristics between the colectomy and non-colectomy groups showed significant differences. In particular, variables associated with a more aggressive course of the disease (i.e., history of steroid dependence and previous exposure to TNFs, increased partial Mayo Clinic score, raised serum CRP level, and reduced albumin level) were more common in patients undergoing colectomy as compared to the control group. This is also in line with data of a recent multicenter retrospective cohort study of adult patients with ASUC admitted to Italian inflammatory bowel disease referral centers from 2005 to 2017, showing that patients avoiding early colectomy were at risk of long-term colectomy, especially if previously exposed to antitumor necrosis factor-α agents, or if rescue therapy during the acute attack was required because of steroid refractoriness [24]. Similarly, the occurrence of colonic dilation was more frequent in the colectomy group as compared to the control group. 

Multivariate analysis showed that a serum albumin level <3 g/dL was a predictor of colectomy. These data fit with those published by Tanaka and colleagues, who showed that a serum albumin level on admission <2.45 g/dL was the most important predictor for early colectomy in patients with moderate to severe UC who did not respond to steroids [18]. Along the same line are the results published by Travis and colleagues prospectively, who monitored 36 clinical, laboratory, and radiographic variables in a cohort of 51 ASUC patients treated with IVCS and rectal hydrocortisone, and 14 with cyclosporine, in order to identify predictors of colectomy and to evaluate the outcome of medical treatment during a one-year follow-up. The authors showed that after 3 days of intensive treatment, a high serum CRP level (>45 mg/L) and stool frequency of >8/day were predictors of colectomy on admission. Additionally, after 7 days’ treatment, patients with >3 stools/day and visible blood had a 60% chance of continuous symptoms and 40% chance of colectomy in the following months [25]. More recently, Le Baut and colleagues created a scoring system to identify patients at low risk and high risk for colectomy within 1 year of hospitalization for ASUC. Among the four independent predictors of colectomy identified, there were a CRP level above 30 mg/L and an albumin level below 30 g/L. These results were confirmed in a validation cohort [13]. In our study, univariate analysis showed that high CRP at admission was a risk factor for colectomy, but multivariate analysis did not confirm such results. Differences in the inclusion criteria (e.g., endoscopic severity of the disease) of the patients as well as in the follow-up duration of these studies can account for the documented discrepancies.

Multivariate analysis also showed that colonic dilation >5.5 cm on admission is a further predictor of colectomy, in line with data of previous studies reporting a high risk of early colectomy in patients with toxic megacolon [26]. Importantly, patients with a serum albumin level ≤2.8 g/dL and colonic dilation > 4 cm on admission had a risk of colectomy 13 times greater than that documented in patients with no risk factor, supporting the view that, despite the induction of clinical remission, some subsets of patients still have a high risk of colectomy. Whether this reflects the inadequacies of medical treatment to dampen mucosal inflammation in this subset of patients remains to be ascertained, even though the demonstration that more than two thirds of the patients underwent surgery within the first year after discharge strongly supports such a hypothesis.

We are aware that our study has some limitations. First, this study was a single-center retrospective study. Second, the small sample size of patients treated with IFX (18 patients) did not allow a comparison between the two populations of treatment. Third, no structured follow-up or treatment algorithm was applied to this study due to its retrospective nature, and data about fecal calprotectin and endoscopic activity were lacking. However, our study has some strengths. This is the first study investigating, in AUC patients responding to medical therapy, the predictive factors of colectomy during such a long follow-up. Moreover, the data were collected from a hospital dataset with a long period, allowing the generation of reproducible results.

## 5. Conclusions

In conclusion, in a real-life population, low albumin levels and colonic dilation at hospital admission are predictors of colectomy in ASUC patients responding to IVCS or IFX. Further prospective studies, even including analysis of objective indicators of ongoing colitis, are needed to confirm the findings of the present report.

## Figures and Tables

**Figure 1 jcm-11-01679-f001:**
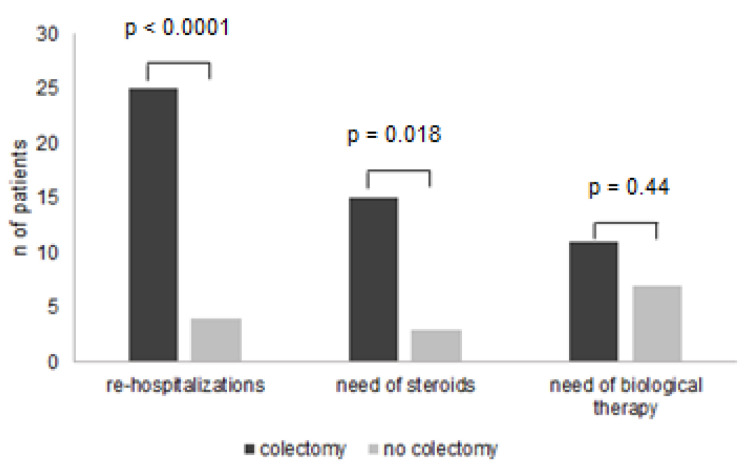
Comparison between colectomy and non-colectomy groups according to the need for re-hospitalization, steroids, and biological therapy during follow-up.

**Figure 2 jcm-11-01679-f002:**
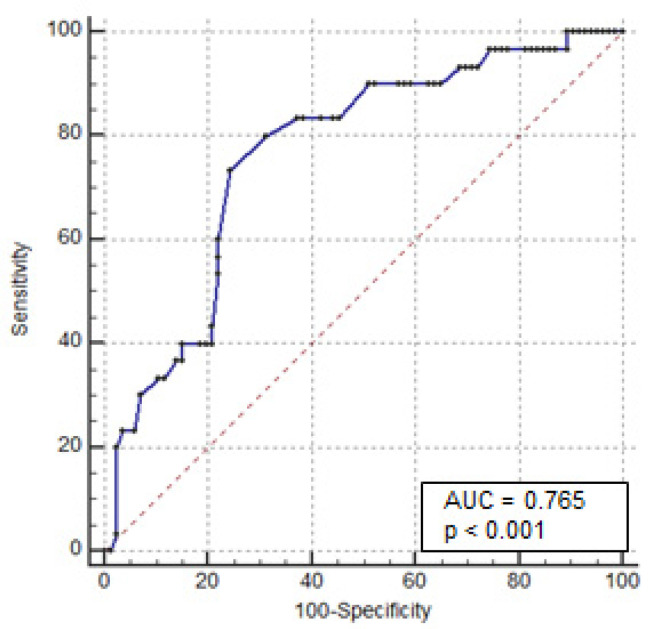
Receiver operating characteristic curves for colectomy based on the serum albumin levels at admission.

**Figure 3 jcm-11-01679-f003:**
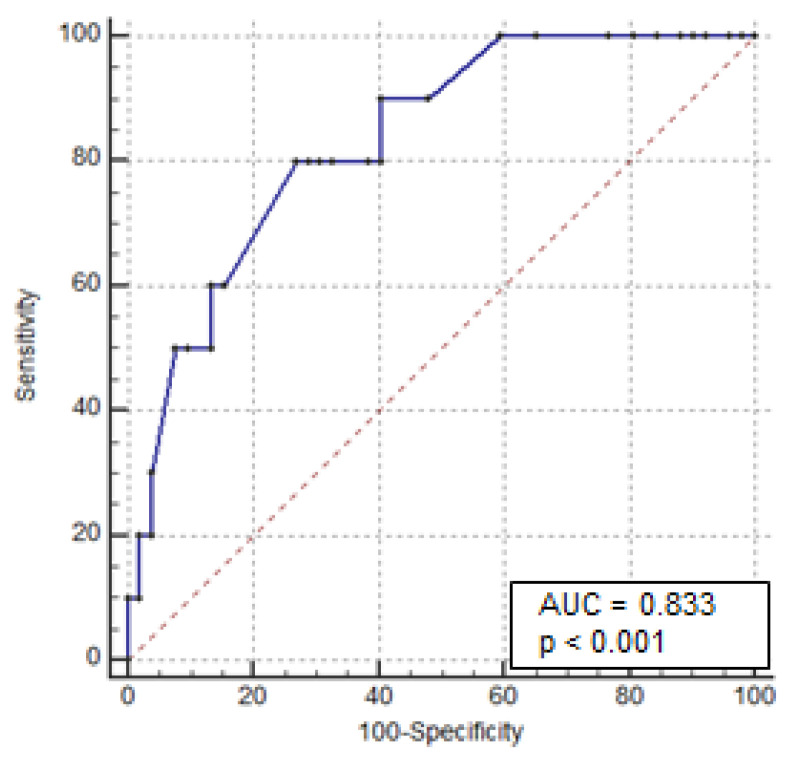
Receiver operating characteristic curves for colectomy based on colonic dilation (X-ray of the abdomen) at admission.

**Figure 4 jcm-11-01679-f004:**
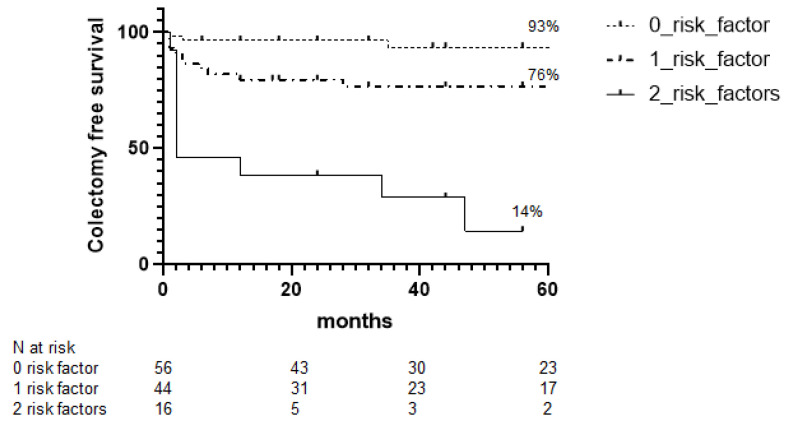
Kaplan–Meier curves of 116 patients with acute severe ulcerative colitis responding to intravenous corticosteroids or infliximab according to risk factors.

**Table 1 jcm-11-01679-t001:** Comparison of clinical parameters and laboratory and radiological data between the non-colectomy and colectomy groups.

Characteristics	Non-Colectomy (87 Patients)	Colectomy (29 Patients)	*p* Value
Female gender, *n* (%)	36 (41%)	21 (72%)	**0.005**
Age, y (median, range)	43 (16–86)	55 (24–85)	0.09
Disease extent, *n* (%)			
E2: left-sided colitis	29 (33%)	9 (31%)	0.9
E3: extensive colitis	58 (67%)	20 (69%)	
Duration of disease			
Months (median, range)	95 (2–504)	168 (2–516)	0.29
Previous anti-TNF therapies, *n* (%)	13 (15%)	10 (34%)	**0.029**
Smoking habits			
Former, *n* (%)	33 (38%)	9 (31%)	0.65
Current, *n* (%)	12 (14%)	3 (10%)	0.76
Steroid dependence			
Yes, *n* (%)	30 (34%)	19 (65%)	**0.04**
Partial Mayo Clinic score (mean ± SD)	6.7 ± 1.03	7.3 ± 1.11	**0.02**
Body mass index (BMI) (median, range)	22.9 (13–32.7)	22.6 (14.3–28.9)	0.75
C-reactive protein (CRP), mg/dL, *n* (%)	38 (0.1–242)	84 (9.5–258)	**0.02**
Albumin, g/dL (median, range)	3.2 (1.6–4.5)	2.8 (1.7–4.3)	**0.04**
Hemoglobin, g/dL (median, range)	11.7 (6.2–16)	11.2 (6.9–13.9)	0.17
Colonic dilation at X-ray, cm (median, range)	3.1 (2.1–6)	4.6 (3.2–6.9)	**0.02**

**Table 2 jcm-11-01679-t002:** Predictive variables associated with colectomy in 116 patients with acute severe ulcerative colitis responding to intravenous corticosteroids or infliximab.

	Univariate Analysis		Multivariate Analysis	
Risk Factors	OR (95% CI)	*p* Value	OR (95% CI)	*p* Value
Female gender	2.914 (1.21–6.97)	**0.01**	5.511 (0.69–43.58)	0.11
Disease duration	1.002 (0.99–1.01)	0.26		-
Extensive colitis (E3)	1.187 (0.48–2.92)	0.71		-
Age at admission	1.021 (0.99–1.05)	0.08		-
Previous anti-TNF exposure	2.808 (1.07–7.34)	**0.04**	1.265 (0.09–18.4)	0.86
Steroid dependence	3.224 (1.36–7.66)	**0.007**	2.724 (0.03–24.52)	0.37
Stool frequency at admission	1.019 (0.93–1.12)	0.7		-
Hemoglobin < 10.5 g/dL	2.102 (0.85–5.22)	0.11		-
C-reactive protein > 30 mg/dL	2.919 (1.01 to 8.43)	**0.034**	1.009 (0.99–1.03)	0.33
Albumin < 3 g/dL	8.741 (3.20–23.8)	**<0.0001**	6.887 (2.08–22.8)	**0.002**
Colonic dilation at X-ray > 5.5 cm	7.581 (2.06–27.87)	**0.0016**	8.468 (1.23–58.3)	**0.03**

## Data Availability

The data that support the findings of this study are available from the corresponding author (G.M.), upon reasonable request.

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
