# Peer review of "Long-Term Risk of Colectomy in Patients with Severe Ulcerative Colitis Responding to Intravenous Corticosteroids or Infliximab"

_jcm, 2022, doi:10.3390/jcm11061679_

Round 1
Reviewer 1 Report
This paper is a very interesting study on long term risk of colectomy in Ulcerative Colitis patients hospitalized for severe flare-up. As the authors correctly pointed out, few data are available in literature about risk factors of long-term colectomy in patients achieving clinical remission with medical therapy. We need more evidences about this topic. However, in my opinion, some points should be reviewed.
MAJOR
- Page 3. Results, paragraph 3.2, line 2
The p-value does not match with that in table 2.
- Page 5-6, paragraph 3.3-figure 4.
There is no match between the text, the 2 risk factors curve and the graph below the curve. Please correct.
Additionally, it would be better to specify how many patients reached 60 months of follow-up.
MINOR
- Page 3. Results, paragraph 3.1
There are some grammatical errors in the last 3 lines. Please correct them.
Lastly, a suggestion: there is a recent Italian work about this topic that you may want to integrate into your discussion.
Festa S., Scribano Maria L, Pugliese D. et al. Long‐term outcomes of acute severe ulcerative colitis in the rescue therapy era: A multicentre cohort study. United European Gastroenterol J. 2021 May; 9(4): 507–516. doi: 10.1177/2050640620977405
Author Response
We would like to thank the reviewer for his/her positive evaluation and helpful suggestions. In response to the specific issues raised by this reviewer:
1. Page 3. Results, paragraph 3.2, line 2. The p-value does not match with that in table 2.
Response: We apologize for the typo. Results section (paragraph 3.2), line 125: the p-value was changed and now matches the one in the table.
2. Page 5-6, paragraph 3.3-figure 4. There is no match between the text, the 2 risk factors curve and the graph below the curve. Please correct. Additionally, it would be better to specify how many patients reached 60 months of follow-up.
Response: Paragraph 3.3, line 156-158: We have corrected the text according to the reviewer suggestion. We have also included the number of patients that reached 60 months of follow-up.
3. Page 3. Results, paragraph 3.1 There are some grammatical errors in the last 3 lines. Please correct them.
Response: Line 120-122: All grammatical errors in the last three lines were corrected.
4. Lastly, a suggestion: there is a recent Italian work about this topic that you may want to integrate into your discussion. Festa S., Scribano Maria L, Pugliese D. et al. Long‐term outcomes of acute severe ulcerative colitis in the rescue therapy era: A multicentre cohort study. United European Gastroenterol J. 2021 May; 9(4): 507–516. doi: 10.1177/2050640620977405
Response: We have revised the discussion section in order to meet the reviewer’s suggestion.
Reviewer 2 Report
An excellent, thoughtfully written manuscript on an important medical topic.
Reference style needs to be corrected, and some awkward sentences are visible. However, this is a problem that can be corrected by the Editorial Office, and it was judged that it does not affect the quality of the entire article.
Author Response
An excellent, thoughtfully written manuscript on an important medical topic. Reference style needs to be corrected, and some awkward sentences are visible. However, this is a problem that can be corrected by the Editorial Office, and it was judged that it does not affect the quality of the entire article
Response: We would like to thank the reviewer for his/her positive evaluation, reference style was corrected according to JCM instructions.
Reviewer 3 Report
The article describes long-term risk and associated risk factors of colectomy of patients with severe ulcerative colitis responding to intravenous corticosteroids or infliximab.
The comments are as follows:
- Abstract: It would be important if the number of words in the abstract allows it to indicate OR and 95% CI
- Introduction: is correctly explained and recent history
- Material and methods:
- Define second line of therapy with IFX
- Were patients treated with cyclosporine?
- ICD 10 codes do not correspond.They're from ICD-9.In this period of time the 2 types of ICD were used successively. Check the codes
- Ethics:Add that the Declaration of Helsinki was followed and if informed consent was required.Were patient data anonymised?
- Describe in more detail the hospital database used fot selection of patients
- Results: In the multivariable analysis, the 2 significant variables have a very wide confidence interval.Interpret this aspect
- Discussion: Would it be possible to reanalyze the data without the 18 IFX patients?
Author Response
We would like to thank the reviewer for his/her positive evaluation and helpful suggestions. In response to the specific issues raised by this reviewer:
Abstract: It would be important if the number of words in the abstract allows it to indicate OR and 95% CI
Response: This data was added in the abstract
Introduction: is correctly explained and recent history
Material and methods:
- Define second line of therapy with IFX
Response: This information was added in the manuscript (line 83-85)
- Were patients treated with cyclosporine?
Response: No patients have been treated with cyclosporine. This piece of information was added to the manuscript.
- ICD 10 codes do not correspond. They're from ICD-9.In this period of time the 2 types of ICD were used successively. Check the codes
Response: the error was corrected in the manuscript (line 68)
- Ethics: Add that the Declaration of Helsinki was followed and if informed consent was required. Were patient data anonymised?
Response: The declaration of Helsinki was added in the manuscript (line 90). We confirm that patient data were anonymised.
- Describe in more detail the hospital database used for selection of patients
Response: An electronic dataset was used for selection of patients. All hospital admissions and respective patient’s charts were included in this database.
Results: In the multivariable analysis, the 2 significant variables have a very wide confidence interval. Interpret this aspect
Response: This is a limit of study due to a small sample size of included patients
Discussion: Would it be possible to reanalyze the data without the 18 IFX patients?
Response: Due to small number of patients treated with IFX, the results did not change even after the removal of this group.
Round 2
Reviewer 1 Report
I appreciated the corrections. No further comments to make.